# Research

analytical chemistry/materials science

$NO_x$ sensor, WO₃ sensing electrode, microstructure, sintering temperature, three-phase boundary

**Author for correspondence:**
Chao Wang
e-mail: wangchao04010502@163.com

This article has been edited by the Royal Society of Chemistry, including the commissioning, peer review process and editorial aspects up to the point of acceptance.

# Effects of WO₃ electrode microstructure on NO₂-sensing properties for a potentiometric sensor

## Bin Yang, Jianzhong Xiao and Chao Wang

State Key Laboratory of Materials Processing and Die and Mold Technology, School of Materials Science and Engineering, Huazhong University of Science and Technology, Wuhan, Hubei 430074, People's Republic of China

(iD) CW, 0000-0001-6581-8098

Planar potentiometric NO₂ sensors based on 8YSZ (8 mol% $Y_2O_3$-doped $ZrO_2$) were prepared with WO₃ sensing electrode material. The various electrode microstructures prepared by different sintering temperatures were characterized by field emission scanning electron microscopy (SEM), and the microstructure influences on the sensors' performances were investigated. The sensor sintered at 800°C, with the most reaction sites, moderate adsorption sites and appropriate electrode thickness, exhibits the highest NO₂ voltage response. While the sensor sintered at 750°C exhibits the lowest NO₂ sensitivity because of the strongest gas-phase catalytic consumption in the WO₃ sensing electrode. Based on the results of SEM characterization and electrochemical impedance spectroscopy tests, the difference in NO₂-sensing performance was attributed to different amounts of electrochemical reaction sites at three-phase boundary, adsorption sites and different degrees of gas-phase catalysis.

## 1. Introduction

The air is continuously polluted by chemical, automobile and petrochemical industries due to rapid industrialization. The poisonous $NO_x$ (NO and NO₂) gases are always blamed for the formation of ozone in the troposphere, the production of acid rains and respiratory problems to humans [1]. In view of the stricter regulations for the emissions from vehicles and other sources of pollutants, the need for a high selectivity, cost-effective and reliable $NO_x$ sensor has become a high priority.

Solid-state potentiometric sensors have been deeply investigated for the detection of $NO_x$. Metal oxide semiconductors, such as WO₃ [2–4], $Cr_2O_3$ [5–7], $SnO_2$ [8] and $LaFeO_3$ [9], are widely used as sensing materials in many potentiometric $NO_x$ sensors with high

detection ability and stability. $WO_3$-based sensor is more promising than other semiconducting metal oxide potentiometric $NO_x$ sensors because of its excellent sensitivity to $NO_x$ without significant sensor signal drift. Lu *et al.* [2] tested various semiconducting metal oxides and found that $WO_3$ could give the best sensing properties to $NO_2$ and NO in the range of 500–700°C. Dutta *et al.* [3] reported a $Y_2O_3$-doped $ZrO_2$ (YSZ)-based sensor with a $WO_3$ sensing electrode ($WO_3$-SE) and a Pt/Au reference electrode; it showed stable, fast and reproducible responses to $NO_2$ at 600–700°C. Di Bartolomeo *et al.* [10] investigated a planar potentiometric sensor with a $WO_3$-SE and a Pt counter electrode on the same side of YSZ, which exhibited a fast and stable response to $NO_2$ and a linear relationship between the response and the $NO_2$ concentrations in a logarithmic scale at 450–700°C. The same linear relationship was also found in a potentiometric tubular sensor with a Pt/$WO_3$ electrode [1]. Yoo *et al.* [11] also prepared a potentiometric sensor with a $WO_3$-SE and a Pt counter electrode on both sides of YSZ and found that it had high sensitivity to even as low as 10 ppm $NO_2$. In view of its excellent sensing performance, $WO_3$ is used as the $NO_2$-sensing electrode material for potentiometric sensor in this work.

For a YSZ-based gas sensor with a metal oxide sensing electrode, electrode microstructure, electrochemical catalytic reaction at the three-phase boundary (TPB) and heterogeneous gas-phase catalytic reaction through the metal oxide electrode are all critical for the gas-sensing performances [12,13]. Large amounts of $NO_2$ molecules may be adsorbed and stored in the sensing electrode, and the electrochemical reactions would take place at the TPB, which provides the diffusion path for $NO_2$ molecules, electrons and $O^{2-}$. So, the well interconnected electrode microstructure between Pt and sensing electrode and compact adhesion of the TPB should be better for $NO_2$ sensing.

For the sensor fabrication processes, many technical parameters could influence the sensing electrode microstructure [14]. As one of the most important parameters, electrode sintering temperature could affect the microstructure as well as the sensing electrode/YSZ interface. Therefore, the sintering procedure plays a significant role in the gas-sensing performance. Based on plenty of earlier publications regarding $WO_3$-SE potentiometric sensors, it could be found that the difference between their sintering procedures is obvious, and they were 700°C for 3 h [2], 750°C for 3 h [3], 800°C for 2 h [7] and 800°C for 10 h [11]. Such big differences suggest that in order to obtain excellent gas-sensing performance, the selection of the sintering procedure for the sensing electrode preparation is very important.

To optimize the fabrication process of the gas sensor and obtain optimal $NO_2$ sensitivity, planar potentiometric sensors with a configuration of Pt/YSZ/(Pt–$WO_3$) sintered at 750, 800 and 850°C for 3 h were investigated in detail. After sintering processes at various temperatures, sensing electrode microstructures and $NO_2$-sensing characteristics of different sensors were tested. The influences of the sintering temperature on the electrode microstructures as well as the sensing properties of $NO_2$ sensors were examined and analysed.

# 2. Experiment

## 2.1. Sensor fabrication

Each $NO_2$ sensor contains one piece of 8 mol% $Y_2O_3$-doped $ZrO_2$ (8YSZ) electrolyte and reference/sensing electrodes located on both surfaces of the electrolyte. The square electrolyte with the dimension of $10 \times 10 \times 0.3$ mm was prepared by the tape casting and sintering process at the temperature of 1500°C for 2 h in air. The reference electrode and Pt collector were screen printed on both surfaces of YSZ electrolyte with Pt slurry (mixed with 1 wt% 8YSZ), dried at 80°C for 2 h and then sintered at 1200°C for 2 h. For the preparation of sensing electrode, commercial tungsten oxide ($WO_3$, 99.9% purity, less than 200 nm, Alfa Aladdin, Shanghai) powder was mixed with 50 wt% binder (5 wt% ethocel, 94 wt% terpineol and 1 wt% Span 80) as the screen-printing slurry. The mixed slurry was screen printed on the Pt collector, dried at 80°C for 2 h and sintered at 750°C, 800°C and 850°C for 3 h each, with a heating and cooling rate of 3°C min$^{-1}$. The schematic of the potentiometric $NO_2$ sensor is shown in figure 1. The surface morphology as well as the thickness of sensing electrode were observed by a field emission scanning electron microscope (SEM, JSM7600F, JEOL) operated at 10 kV.

## 2.2. Evaluation of sensing properties

Sensor performance was tested in a gas flow apparatus under various gas environments controlled by mass flow controllers (MPA-80, Beijing Seven Star Electronics Company). The gas flow apparatus was connected to a quartz tube equipped with a furnace operating at temperatures in the range of 500–

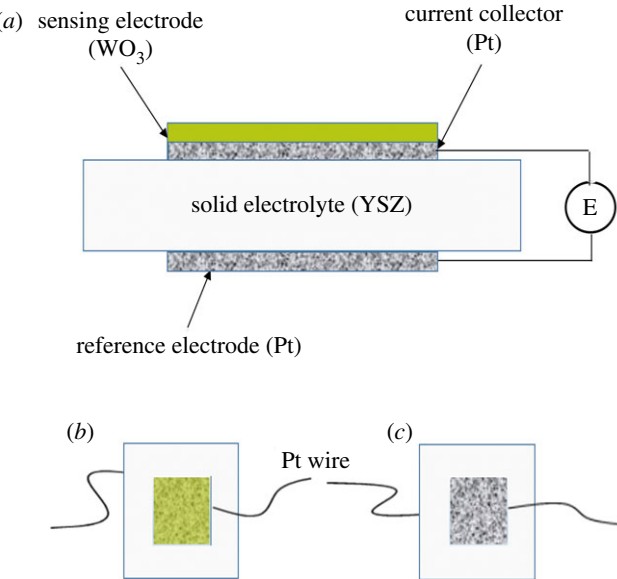

**Figure 1.** (a) Schematic of the fabricated sensor; (b) top view of the sensor and (c) bottom view of the sensor.

600°C. The gas environment consisted of a changing $NO_2$ concentration varying from 100 to 500 ppm and the base gas (10 vol.% $O_2 + N_2$ balance), at a total flow rate of $0.2\,l\,min^{-1}$. Both electrodes of the sensor were exposed to the same gas atmosphere. The voltage difference $\Delta V$ (open circuit potential) between the sensing and reference electrodes was monitored and recorded by an electrochemical work station (Versa STAT 3, Princeton, USA). For the voltage measurements, the sensing electrode was connected to the positive terminal of the electrochemical work station, and the reference electrode was connected to the negative terminal.

Electrochemical impedance spectroscopy (EIS) was conducted with the electrochemical work station in the frequency range of 0.1 Hz to 1 MHz, with 10 mV exciting voltage at 550°C.

# 3. Results and discussion

## 3.1. Morphology of $WO_3$ sensing electrode

SEM micrographs of the cross sections of $WO_3$-SEs are shown in figure 2. It is seen that $WO_3$-SEs sintered at various temperatures have different thicknesses and there is some variability in the thickness throughout the same electrode. So, we took several measurements across the cross section of the electrode from large areas and calculated statistical average thickness for each electrode. The average thicknesses of the electrodes are 35, 25 and 10 μm. These results show that the thickness of $WO_3$-SE decreases seriously with increasing of sintering temperature. Figure 3 shows surface micrographs of $WO_3$-SE sintered at different temperatures. It can be seen from figure 3a,b that the Pt collector of the sensors sintered at 750°C and 800°C is fully covered by $WO_3$ layer. Moreover, in figure 3c, the Pt collector of the sensor sintered at 850°C is not fully covered by $WO_3$, and some part of the Pt collector is exposed to gas.

Figure 4 shows the surface morphology of each $WO_3$-SE sintered at different temperatures. It is observed that the average grain size evaluated in large areas is about 2 μm, for each $WO_3$-SE. For the sample sintered at 750°C, $WO_3$ particles are relatively dispersed without obvious sintering phenomenon. But for the sample sintered at 800°C, proper sintering necks and a network structure were formed between the $WO_3$ particles. For the 850°C-sintered sample, a serious sintering phenomenon was generated due to the high temperature, which led to the destruction of electrode network structure.

## 3.2. Evaluation of sensing characteristics

Figure 5 shows the voltage responses of the sensors sintered at 750, 800 and 850°C to different $NO_2$ concentrations (10 vol.% $O_2 + N_2$ balance) at 500–600°C. The voltages in these plots are an average of the response values at each concentration. The average voltage was calculated from three values measured at each $NO_2$ concentration of 100, 200, 400 and 500 ppm. For each operating temperature and $NO_2$

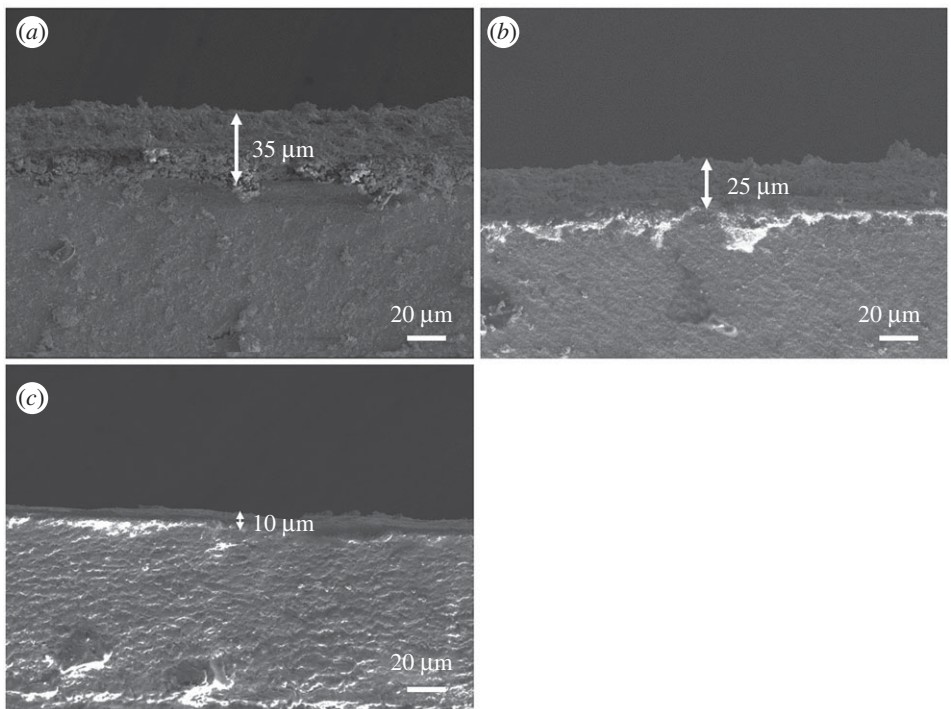

**Figure 2.** SEM micrographs of the cross sections of WO$_3$-SE sintered at different temperature. (*a*) 750°C, (*b*) 800°C and (*c*) 850°C.

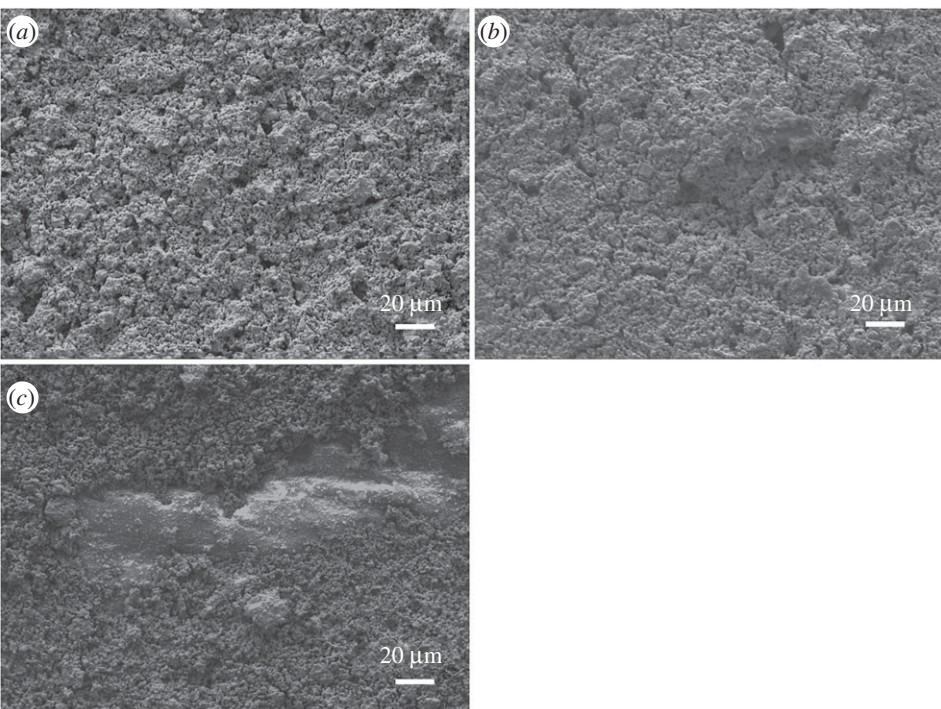

**Figure 3.** Surface micrographs of each WO$_3$-SE sintered at different temperatures. (*a*) 750°C, (*b*) 800°C and (*c*) 850°C.

concentration, the lowest $\Delta V$ was always obtained from the sensor sintered at 750°C, and the highest value was obtained from the sample sintered at 800°C. The sensor sintered at 800°C had the fastest response and recovery, while the sensor sintered at 750°C showed the slowest rates. It could also be seen that NO$_2$ sensitivity decreases significantly for each sensor as operating temperature increases.

Figure 6 displays the dependence of response values on the logarithm scale of NO$_2$ concentration for each sensor. It can be seen that in each case, the voltage response increases linearly with an increasing NO$_2$ concentration on a logarithmic scale. Such a linear variation is typical for a mixed potential-type gas sensor [15,16].

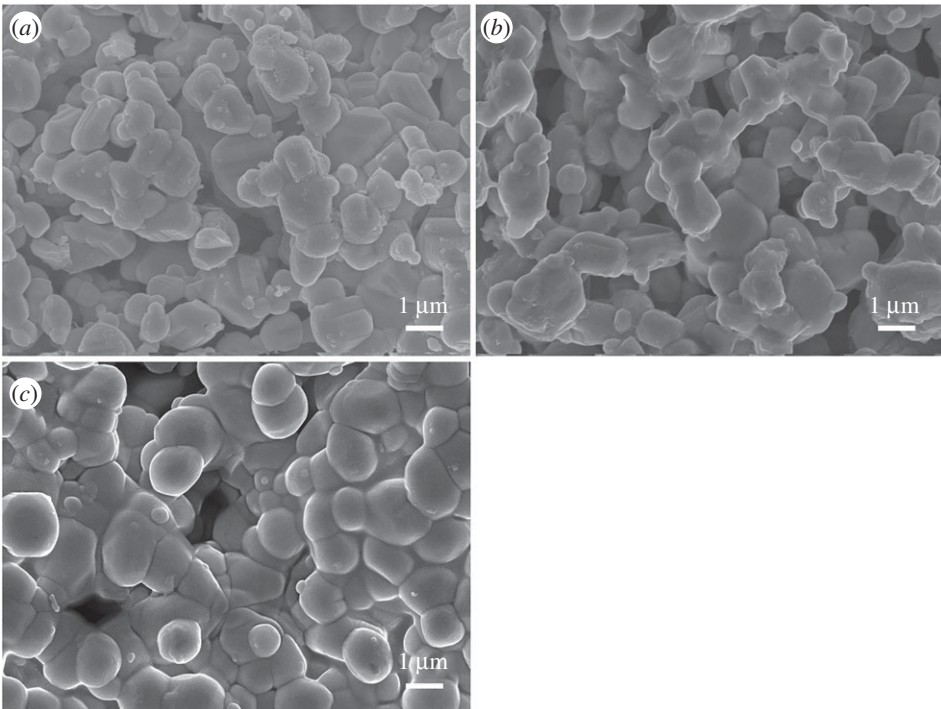

**Figure 4.** SEM images of each WO$_3$-SE sintered at different temperatures. (*a*) 750°C, (*b*) 800°C and (*c*) 850°C.

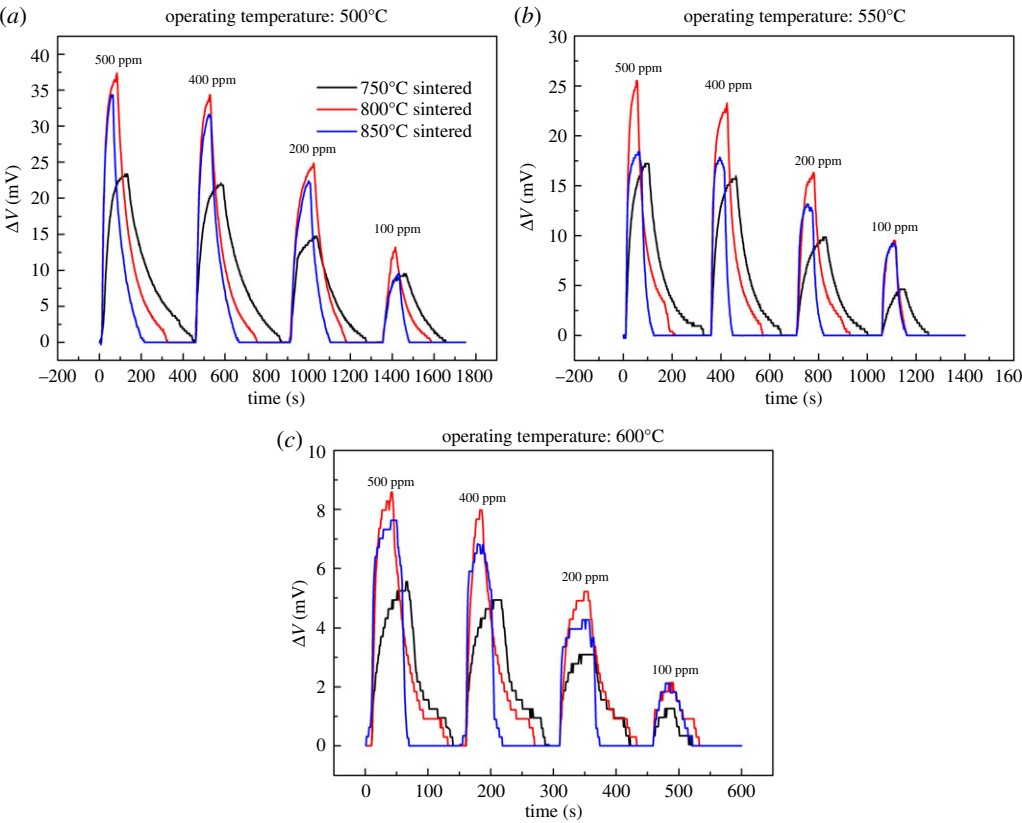

**Figure 5.** NO$_2$ voltage response versus time at (*a*) 500°C, (*b*) 550°C and (*c*) 600°C.

## 3.3. Effect of electrode microstructure on sensing performance

The sensing mechanism of the YSZ-based potentiometric NO$_2$ sensor can be basically explained by the mixed potential mechanism existing at the TPB of WO$_3$ (Pt), YSZ and gas [17]. The mixed potential

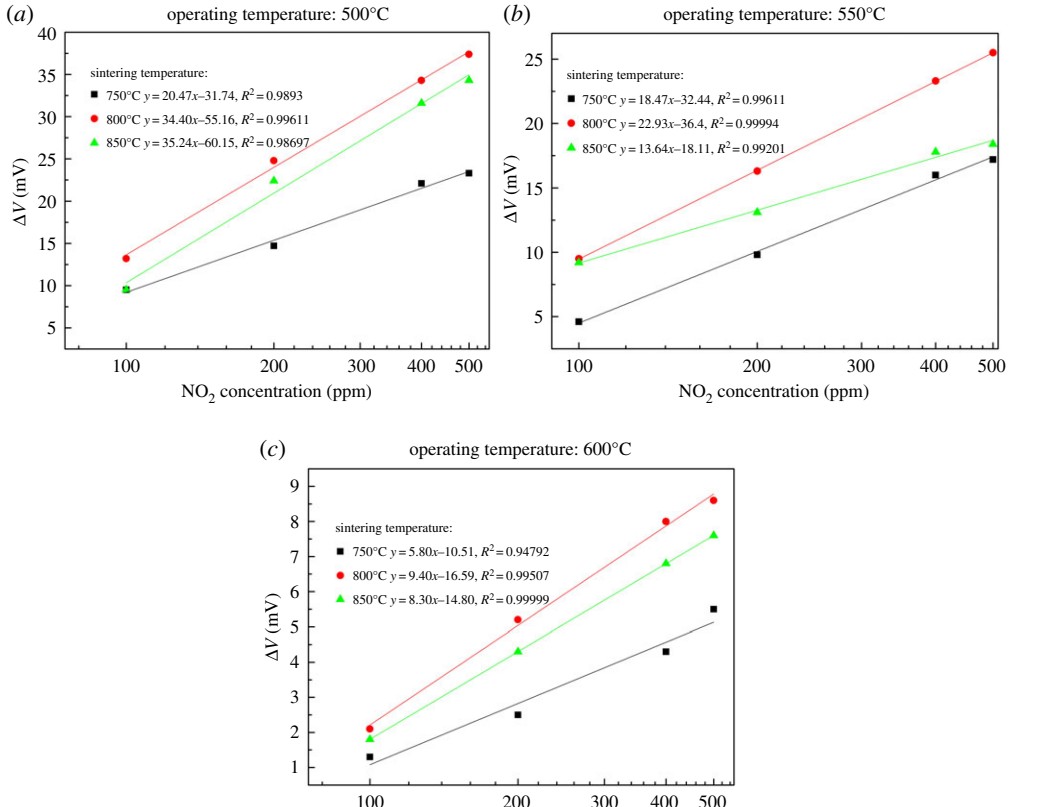

**Figure 6.** Voltage response versus log(NO$_2$ concentration) for each WO$_3$-SE at (a) 500°C, (b) 550°C and (c) 600°C.

appears when the electrochemical anode reaction (3.1) and the cathode reaction (3.2) proceed simultaneously with an equal rate at the TPB:

$$O^{2-} \leftrightarrows 1/2O_2(g) + 2e^- \tag{3.1}$$

$$NO_2(g) + 2e^- \leftrightarrows NO(g) + O^{2-} \tag{3.2}$$

When a larger number of electrochemical reaction sites are present at the TPB, the sensor could exhibit faster response/recovery and a higher voltage response; when the number of reaction sites is significantly lower, the response/recovery rates and sensitivity could be lower. The number of reaction sites at the TPB is greatly affected by the microstructure of the sensing electrode. In general, the sensing electrode with fine particles and three-dimensional network structure possesses more gas reaction sites, and the electrode with higher porosity shows faster response and recovery. As can be seen from figure 4, although the particles of WO$_3$-SE sintered at 750°C were uniformly distributed, the sensing electrode did not present a porous network structure compared with the sample sintered at 800°C, so it was not conducive to the diffusion of gas molecules to the TPB and the electrode reactions. Therefore, the 800°C-sintered sensor exhibited higher NO$_2$ sensitivity and response/recovery rates.

However, the sensing properties of the 850°C-sintered sample did not continue to increase with rising temperature, and the response and recovery rates also slowed down. Combining with the microstructure analysis of sensing electrode, it is thought that this is due to the serious sintering phenomenon caused by the higher sintering temperature. The sintering necks among particles decreased the diffusion channels and reduced reaction sites, and it eventually led to the decline of sensor performance.

Besides reaction sites, the number of adsorption sites on the sensing electrode also plays an important role in sensor performances. The more adsorption sites lead to the higher gas concentration at the TPB; it enhances the intensity of the electrochemical reactions as well as the NO$_2$ sensitivity of the sensor. WO$_3$ is an n-type semiconducting oxide, and when WO$_3$-SE is exposed to NO$_2$, NO$_2$ molecules can be adsorbed directly in WO$_3$-SE [18–20]. As WO$_3$ is a low temperature sublimation material [21,22], it begins to sublimate at 650°C, and the sublimation becomes more serious with an increasing sintering temperature. As can be seen from figure 3, different sintering temperatures led to different degrees of WO$_3$ sublimation, which resulted in different micro-morphology of sensing electrodes. Specifically, the

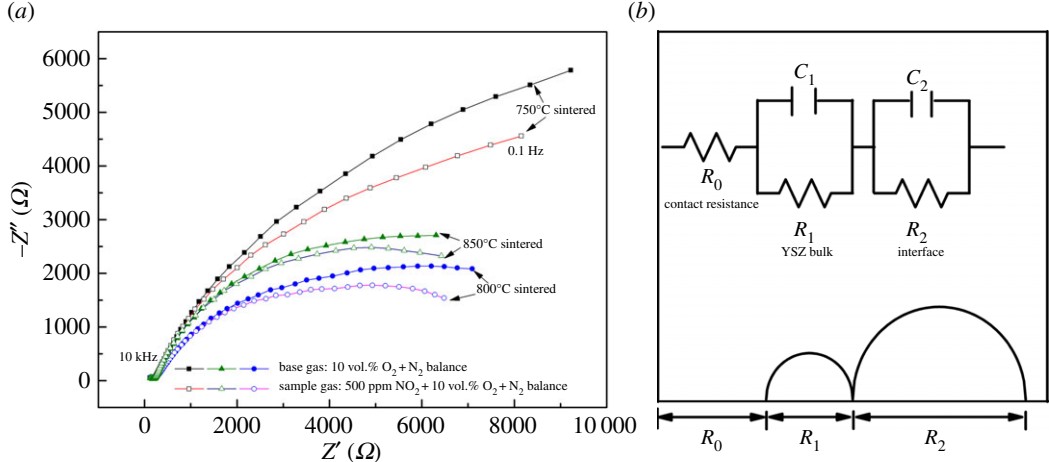

**Figure 7.** (a) Electrochemical impedance spectroscopy in the base gas and sample gas at 550°C for the sensors sintered at different temperatures. (b) Nyquist plot of an ideal equivalent circuit for YSZ-based potentiometric sensor.

electrodes sintered at 750 and 800°C show a complete structure. But, for the 850°C-sintered sample, part of the Pt collector was exposed to gas due to the $WO_3$ sublimation, which destroyed the integrity of the electrode structure and thus reduced the adsorption sites and reaction sites of the sensing electrode for $NO_2$. Therefore, from the perspective of the gas adsorption sites, the 850°C-sintered sample should have the lowest $NO_2$ sensitivity, but it is inconsistent with the results in figure 5; the sensitivity of the 850°C-sintered sample is higher than that of the 750°C-sintered sample.

It is reported that the thickness of sensing electrode of gas sensor has a great influence on sensor performance [23,24]. The thicker sensing electrode layer leads to more serious gas-phase catalytic reaction, which causes the target gas reaching the TPB tending to be in equilibrium, reduces the concentration of $NO_2$ and decreases the sensitivity of the sensor. For the sample sintered at 800°C, the $WO_3$-SE layer thickness was significantly less than that of the 750°C-sintered sample; therefore, the gas-phase consumption effect was relatively low, resulting in a higher concentration of $NO_2$ at the TPB, which eventually resulted in a greater sensitivity. For the $WO_3$-SEs sintered at 750 and 850°C, although the higher sintering temperature led to the incomplete electrode structure and the reduced $NO_2$ adsorption sites, the thinner electrode layer slowed down the gas-phase consumption of $NO_2$ to the greatest degree. These two factors with the opposite effect eventually led to the higher $NO_2$ sensitivity of the 850°C-sintered sample.

To further understand the effect of sensing electrode microstructure on gas sensitivity, the electrochemical impedance spectra of the sensors sintered at different temperatures were measured between 0.1 Hz and 1 MHz at 550°C in the base gas and sample gas (500 ppm $NO_2$); the Nyquist plots are shown in figure 7a. It could be seen that there is a smaller semicircle which is not obvious in the high frequency range and a bigger semicircle in the low frequency range. The shape and size of the small arc at the high frequency region are similar for the sensors sintered at different temperatures, either in the base gas or the sample gas. But there is a significant shrinkage towards the real $Z'$-axis for the large arc at low frequency when $NO_2$ is injected.

A probable equivalent circuit could be given for the potentiometric gas sensor based on some earlier reports, as shown in figure 7b. Here, $R_0$ represents the resistance caused by electric connections in the testing device and a gas sensor. The Voigt element at high frequencies ($R_1C_1$) corresponds to the bulk resistance and the capacity of the YSZ electrolyte due to the uniformity of the shape and size for different samples, whereas the Voigt element at low frequencies ($R_2C_2$) corresponds to interface resistance and capacity for electrochemical reactions at the TPB [25–27]. The resistance values of the equivalent circuit were calculated with Zview software and are shown in table 1. It could be seen that $R_0$ and $R_1$ do not change obviously, but $R_2$ has a great decline with the gas exchange from base gas to sample gas. And as the sintering temperature increases, the interface resistance also changes obviously, which is the lowest for the sample sintered at 800°C but highest at 750°C. The magnitude of the interface resistance usually reflects the degree of electrochemical reactions at the TPB: a smaller resistance means a more intense electrode reaction [26,28]. Here, because the $NO_2$ concentration in the sample gas is the same, so the intensity of electrode reactions is mainly affected by the microstructure and $WO_3$-SE thickness introduced by the different sintering temperatures. In this work, the sample

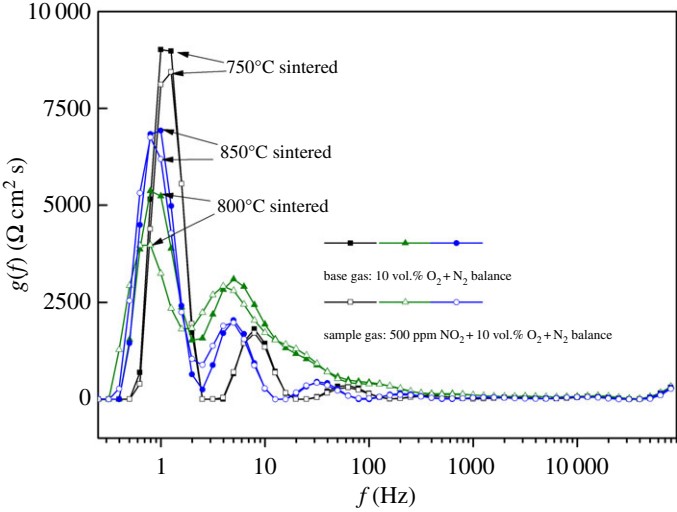

**Figure 8.** DRT analysis deconvoluted from EIS results of figure 7a.

**Table 1.** Resistance values of equivalent circuit in the base gas and sample gas at 550°C for the sensors sintered at different temperatures.

| sintering temperature (°C) | $R_0$ (Ω) | | $R_1$ (Ω) | | $R_2$ (Ω) | |
|---|---|---|---|---|---|---|
| | base gas | sample gas | base gas | sample gas | base gas | sample gas |
| 750 | 56 | 49 | 162 | 152 | 20 958 | 14 683 |
| 800 | 50 | 47 | 222 | 217 | 9488 | 8065 |
| 850 | 55 | 46 | 154 | 146 | 9640 | 8722 |

sintered at 800°C has appropriate adsorption sites, most reaction sites and moderate electrode thickness, and shows the best $NO_2$ sensitivity.

The distribution of relaxation times (DRT) is also a useful method for deconvoluting EIS data and has been successfully used to identify the reaction mechanisms in solid oxide fuel cell and battery materials [29–32]. It can display impedance data as a distribution of time constants, which can be easier to interpret which physical process is the determining factor for the entire electrochemical process. We tried to deconvolute the EIS results with the DRT method and the results are shown in figure 8. It could be found that each curve has a double peak between 0.1 and 10 Hz; there is roughly the same position of the peaks for the three samples either in the base gas or sample gas. Normally, the electrode reaction is the critical step in the entire electrochemical processes in the low frequency range. So, the peaks appearing in the low frequency range and the similar peak positions should be caused by the same process of electrode reactions.

It could also be seen that under the same concentration of $NO_2$, the smallest peak area is from the 800°C-sintered sample, and the largest one is from the 750°C-sintered sample. The area of the peak represents the polarization resistance of the electrode reaction process [29]; the smaller the area, the lower the electrode reaction resistance and the more intense the electrode reaction. Perhaps, the 800°C-sintered sample had the optimal morphology and electrode thickness, so that it had the strongest electrode reaction and finally obtained the maximum response signal.

In conclusion, based on the analysis of different factors for sensor performance, in order to obtain the best sensing performance, it is necessary to adopt the proper sintering procedure and to get the optimum sensing electrode microstructure.

# 4. Conclusion

The mixed potential-type sensors using the YSZ electrolyte and $WO_3$-SE sintered at different temperatures were fabricated; the sensing electrode microstructure and $NO_2$-sensing characteristics

were examined. It was shown that the $NO_2$-sensing performance was strongly dependent on the sensing electrode microstructure. The change of electrode microstructure could influence the reaction sites, adsorption sites and gas-phase catalysis, and then affect the sensing performance of the sensor. The 800°C-sintered sensor with moderate adsorption sites in $WO_3$-SE, the most reaction sites at TPB and moderate electrode thickness had the highest voltage response and fastest response/recovery. However, for the samples sintered at 750°C and 850°C, due to different amounts of electrochemical reaction sites at TPB, adsorption sites and different degrees of gas-phase catalytic consumption, the 850°C-sintered sample presented the higher $NO_2$ sensitivity.

Data accessibility. This article has no additional data.

Authors' contributions. B.Y. conceived of and designed the experiments, analysed the data and made contribution in preparation of the manuscript. C.W. analysed the data and did the final editing of the manuscript. J.z.X. helped to solve the problems related to the accuracy or integrity of any part of the work which are appropriately investigated and resolved. All authors gave their final approval for submission of the manuscript.

Competing interests. We declare we have no competing interests.

Funding. We received no funding for this study.

Acknowledgements. The authors thank the Analytical and Testing Center of the Huazhong University of Science and Technology for its hard work in this article's XRD and SEM tests.

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
