## [Reviewer comments · Royal Society Open Science]

Review History

RSOS-190526.R0 (Original submission)

Review form: Reviewer 1

Is the manuscript scientifically sound in its present form?

Yes

Are the interpretations and conclusions justified by the results?

Yes

Is the language acceptable?

Yes

Is it clear how to access all supporting data?

Yes

Do you have any ethical concerns with this paper?

No

Have you any concerns about statistical analyses in this paper?

No

Recommendation?

Accept with minor revision (please list in comments)

Comments to the Author(s)

What is the innovation of the manuscript? It should be clearly stated.

Review form: Reviewer 2 (Ho Yee Hui)**Is the manuscript scientifically sound in its present form?**

Yes

Are the interpretations and conclusions justified by the results?

Yes

Is the language acceptable?

Yes

Is it clear how to access all supporting data?

Not Applicable

Do you have any ethical concerns with this paper?

No

Have you any concerns about statistical analyses in this paper?

Yes

Recommendation?

Major revision is needed (please make suggestions in comments)

Comments to the Author(s)

The authors report on a sintering temperature effect on a type of material for NO₂ sensing application. The NO₂ sensor is composed of YSZ with WO₃ as the electrode for the sensing measurements. The novelty of this study comes from the connection between the temperature effect and material system combination (YSZ + WO₃ + NO₂). SEM is used to understand the surface morphology along with standard sensing and EIS measurements. I do appreciate that authors connect the fundamental diffusion and reaction mechanisms to the surface morphology and voltage changes. Overall, the work has reached a decent level of novelty, but more understanding about the data sets should be included in the content. As such, I recommend that this work be published in Royal Society Open Science after the following major revisions:

1. The voltage responses are similar between 800 and 850C sintered data at the operating temperature 500C in Figure 5a. However, the responses are lage between 800 and 850C sintered data at the operating temperature 550 and 600C in Figure 5b and 5c. Can authors explain more in the manuscript?
2. All voltage responses data in Figure 6a-6c have the non-zero y-intercept when NO₂ concentration is 0 ppm. Can authors elaborate more about this observation here?
3. Voltage response is increased proportionally with NO₂ concentration in Figure 6a-6c except the 850C sintering temperature data points in Figure 6b. Can authors discuss more about this observation here?

4. Authors discuss that the thickness of the electrode changes with the sintering temperature. This thickness effect is competing with the microstructure formation as a function of the sintering temperature. Which effect is more impactful on the sensing capability based on two different sets of data here in Figure 5 and 6?
5. Authors should combine the Figure 7 and 8 together for more efficient usage of manuscript layout.

Review form: Reviewer 3

Is the manuscript scientifically sound in its present form?

No

Are the interpretations and conclusions justified by the results?

No

Is the language acceptable?

No

Is it clear how to access all supporting data?

Yes

Do you have any ethical concerns with this paper?

No

Have you any concerns about statistical analyses in this paper?

Yes

Recommendation?

Major revision is needed (please make suggestions in comments)

Comments to the Author(s)

These authors used deposited WO₃ on YSZ and used it as a NO₂ sensor. The authors saw a dependence on the sintering conditions with 800 C delivering the best performance. They attribute the improved performance to the morphology.

- 1) The authored should have used single crystal substrates to test if the WO₃ morphology depends on the substrate surface, termination, morphology, and conditions.
- 2) The authors should provide a compelling explanation as to why sintering yields different thicknesses. Now it seems to violate the most basic mass conservation.
- 3) The morphology cannot be obtained or even cursorily analyzed with Fig. 3 as in Fig. 3 I only see qualitative differences. AFM or profilometry would be needed.
- 4) I am not sure why the authors used so few points in Fig. 6 they seem to have quite a bit of scattering. Experiments should be repeated on other samples, and the fitting curve and R² are not very meaningful. Also, there are multiple figs 6.
- 5) It is very unclear how (if at all) the circuit in Fig. 8 fits with Fig. 7. The authors are recommended to carry out DRT analysis (see <https://www.bio-logic.net/wp-content/uploads/AN60.pdf> and key references therein)
- 6) The quality of the figures needs improvement as the SEM is not aligned and the fonts are elongated. The schematics are visually unappealing. Also, the labeling of the figures should be formatted consistently.

7) The writing should be improved, and the article should be checked by a native speaker or a professional editor.

Decision letter (RSOS-190526.R0)

10-May-2019

Dear Dr Wang:

Title: Effects of WO₃ electrode microstructure on NO₂ sensing properties for potentiometric sensor

Manuscript ID: RSOS-190526

The editor assigned to your manuscript has now received comments from reviewers. We would like you to revise your paper in accordance with the referee and Subject Editor suggestions which can be found below (not including confidential reports to the Editor). Please note this decision does not guarantee eventual acceptance.

Please submit your revised paper before 02-Jun-2019. Please note that the revision deadline will expire at 00.00am on this date. If we do not hear from you within this time then it will be assumed that the paper has been withdrawn. In exceptional circumstances, extensions may be possible if agreed with the Editorial Office in advance. We do not allow multiple rounds of revision so we urge you to make every effort to fully address all of the comments at this stage. If deemed necessary by the Editors, your manuscript will be sent back to one or more of the original reviewers for assessment. If the original reviewers are not available we may invite new reviewers.

Royal Society of Chemistry

Thomas Graham House
Science Park, Milton Road
Cambridge, CB4 0WF
Royal Society Open Science - Chemistry Editorial Office

RSC Associate Editor:
Comments to the Author:
(There are no comments.)

RSC Subject Editor:
Comments to the Author:
(There are no comments.)

Reviewers' Comments to Author:
Reviewer: 1

Comments to the Author(s)
What is the innovation of the manuscript? It should be clearly stated.

Reviewer: 2

Comments to the Author(s)
The authors report on a sintering temperature effect on a type of material for NO₂ sensing application. The NO₂ sensor is composed of YSZ with WO₃ as the electrode for the sensing measurements. The novelty of this study comes from the connection between the temperature effect and material system combination (YSZ + WO₃ + NO₂). SEM is used to understand the surface morphology along with standard sensing and EIS measurements. I do appreciate that authors connect the fundamental diffusion and reaction mechanisms to the surface morphology and voltage changes. Overall, the work has reached a decent level of novelty, but more understanding about the data sets should be included in the content. As such, I recommend that this work be published in Royal Society Open Science after the following major revisions:

1. The voltage responses are similar between 800 and 850C sintered data at the operating temperature 500C in Figure 5a. However, the responses are lage between 800 and 850C sintered data at the operating temperature 550 and 600C in Figure 5b and 5c. Can authors explain more in the manuscript?
2. All voltage responses data in Figure 6a-6c have the non-zero y-intercept when NO₂ concentration is 0 ppm. Can authors elaborate more about this observation here?
3. Voltage response is increased proportionally with NO₂ concentration in Figure 6a-6c except the 850C sintering temperature data points in Figure 6b. Can authors discuss more about this observation here?
4. Authors discuss that the thickness of the electrode changes with the sintering temperature. This thickness effect is competing with the microstructure formation as a function of the sintering

temperature. Which effect is more impactful on the sensing capability based on two different sets of data here in Figure 5 and 6?

5. Authors should combine the Figure 7 and 8 together for more efficient usage of manuscript layout.

Reviewer: 3

Comments to the Author(s)

These authors used deposited WO₃ on YSZ and used it as a NO₂ sensor. The authors saw a dependence on the sintering conditions with 800 C delivering the best performance. They attribute the improved performance to the morphology.

- 1) The authored should have used single crystal substrates to test if the WO₃ morphology depends on the substrate surface, termination, morphology, and conditions.
- 2) The authors should provide a compelling explanation as to why sintering yields different thicknesses. Now it seems to violate the most basic mass conservation.
- 3) The morphology cannot be obtained or even cursorily analyzed with Fig. 3 as in Fig. 3 I only see qualitative differences. AFM or profilometry would be needed.
- 4) I am not sure why the authors used so few points in Fig. 6 they seem to have quite a bit of scattering. Experiments should be repeated on other samples, and the fitting curve and R² are not very meaningful. Also, there are multiple figs 6.
- 5) It is very unclear how (if at all) the circuit in Fig. 8 fits with Fig. 7. The authors are recommended to carry out DRT analysis (see <https://www.bio-logic.net/wp-content/uploads/AN60.pdf> and key references therein)
- 6) The quality of the figures needs improvement as the SEM is not aligned and the fonts are elongated. The schematics are visually unappealing. Also, the labeling of the figures should be formatted consistently.
- 7) The writing should be improved, and the article should be checked by a native speaker or a professional editor.

Author's Response to Decision Letter for (RSOS-190526.R0)

See Appendix A.

Decision letter (RSOS-190526.R1)

04-Jun-2019

Dear Dr Wang:

Title: Effects of WO₃ electrode microstructure on NO₂ sensing properties for potentiometric sensor

Manuscript ID: RSOS-190526.R1

It is a pleasure to accept your manuscript in its current form for publication in Royal Society Open Science. The chemistry content of Royal Society Open Science is published in collaboration with the Royal Society of Chemistry.

RSC Associate Editor
Comments to the Author:
(There are no comments.)

Reviewer(s)' Comments to Author:

Appendix A

Reviewer: 1

What is the innovation of the manuscript? It should be clearly stated.

Answer: The sintering temperature affects the microstructure of the sensing electrode (SE), which in turn affects the electrical properties of the gas sensor. Generally, in order to reduce gas consumption caused by gas phase catalysis during gas diffusion in the SE, the SE should be controlled at a suitable thickness. But for volatile material WO_3 , in order to avoid the damage of the integrity of WO_3 -SE caused by volatilization of electrode material, and ensure the proper electrode thickness, it is particularly important to control the sintering temperature. This paper innovatively combines the SEM results, NO_2 sensing performance and EIS results to reveal how the morphology of the SE affects the sensing performance by affecting the reaction sites, adsorption sites, and gas phase reaction consumption.

Reviewer: 2

1. The voltage responses are similar between 800 and 850°C sintered data at the operating temperature 500°C in Figure 5a. However, the responses are large between 800 and 850°C sintered data at the operating temperature 550 and 600°C in Figure 5b and 5c. Can authors explain more in the manuscript?

Answer: Compared with the 800 °C sintered WO_3 -SE, the electrode morphology of 850 °C sintered WO_3 -SE is incomplete, resulting in fewer adsorption sites for NO_2 molecules. At a lower operating temperature of 500 °C, the difference in response signal caused by adsorption sites difference is small. But as the operating temperature increases, the gas desorption effect enhances, and the signal gap caused by the adsorption sites difference becomes larger. Therefore, we could see a similar signal at 500 °C (5a) and a significant signal difference at 550 °C (5b) and 600 °C (5c).

2. All voltage responses data in Figure 6a-6c have the non-zero y-intercept when NO_2 concentration is 0 ppm. Can authors elaborate more about this observation here?

Answer: In this study, the sensor consists of a piece of YSZ solid electrolyte and a pair of symmetrical Pt electrodes screen printed on both sides of YSZ, and one of Pt electrode is coated with a layer of WO_3 sensing material. In the base gas (0 ppm NO_2), the response signal is derived from the oxygen potential difference in the vicinity of

the two Pt electrodes. Since the printing process is difficult to ensure the perfect symmetry of Pt electrodes on both sides, and the layer of sensing material coated on one Pt hinders the transmission of oxygen to some extent, that finally resulting in a non-zero response signal due to the different oxygen potential on the both Pt electrodes. So non-zero y-intercept could be found when NO₂ concentration is 0 ppm.

3. Voltage response is increased proportionally with NO₂ concentration in Figure 6a-6c except the 850°C sintering temperature data points in Figure 6b. Can authors discuss more about this observation here?

Answer: In Section 3.2 of the experimental introduction of this paper, it has been mentioned that the response signal of each concentration is the average value from the three measurement results. It may be due to the experimental error and the insufficient samples, the relationship between the response signal and the logarithm of the NO₂ concentration does not have a good linear relationship, and this phenomenon is very common in the potentiometric sensor literature. Sensors of this structure generally follow the mixed-potential mechanism, so we usually use a linear fitting method when analyzing the potential signal with the logarithm of gas concentration.

4. Authors discuss that the thickness of the electrode changes with the sintering temperature. This thickness effect is competing with the microstructure formation as a function of the sintering temperature. Which effect is more impactful on the sensing capability based on two different sets of data here in Figure 5 and 6?

Answer: The difference in sintering temperature will result in the difference in the microstructure and the thickness of the electrode. The microstructure of the electrode will affect the adsorption sites and the reaction sites, and the thickness will affect the gas phase catalytic reaction consumption of the target gas and affect the gas concentration reaching the reaction sites at the TPB. Combined with the microstructure observation of SEM and the test signal results of Fig. 5 and 6, it is considered that the thickness effect is more impactful on sensing performance.

5. Authors should combine the Figure 7 and 8 together for more efficient usage of manuscript layout.

Answer: Thanks for your advice. I will modify my manuscript and combine the two figures.

Reviewer: 3

1) The author should have used single crystal substrates to test if the WO₃ morphology depends on the substrate surface, termination, morphology, and conditions.

Answer: A single crystal substrate is generally used for a resistive type gas sensor, which has a semiconductor metal oxide printed on the substrate. The sintering temperature of the resistive sensor is generally less than 500 °C, and the test temperature is below 300 °C. But for the potentiometric sensor, the YSZ substrate is not only a structural support, but also has the function of an oxygen ion conductor. In this work, the YSZ is an isotropic polycrystalline material prepared by the same tape-casting and sintering procedure. Therefore, all YSZ substrates have the same properties and surface conditions. The sensing electrode WO₃ studied in this paper are prepared by the same batch of printing, except the different electrode sintering temperature which could not influence the YSZ surface conditions, so the different electrode morphology comes from the sintering temperature difference, not from the substrate difference.

2) The authors should provide a compelling explanation as to why sintering yields different thicknesses. Now it seems to violate the most basic mass conservation.

Answer: In the case where the sintering temperature is lower than the melting temperature of the electrode material, the sintering process does not cause a significant change in the electrode thickness. However, the sensing material WO₃ used in this paper is an easily sublimation material, and it begins to sublime above 650 °C which is mentioned in Section 3.3. The higher the temperature, the more obvious the sublimation. So the difference in electrode thickness is caused by varying degrees of sublimation caused by different sintering temperatures.

3) The morphology cannot be obtained or even cursorily analyzed with Fig. 3. As in Fig. 3, I only see qualitative differences. AFM or profilometry would be needed.

Answer: The AFM equipment test mode of the school test center is contact type. The WO₃ material used in the paper is easily destroyed due to its crisp structure. The test center does not recommend us to use this equipment. In fact, the surface morphology difference of the sintered electrode at different temperatures can be seen by the SEM

analysis. As the sintering temperature increased, the sensing electrode became crispy and porous, and there is a significant Pt electrode exposure phenomenon caused by WO_3 sublimation at the sintering temperature of 850 °C.

4) I am not sure why the authors used so few points in Fig. 6 they seem to have quite a bit of scattering. Experiments should be repeated on other samples, and the fitting curve and R^2 are not very meaningful. Also, there are multiple figs 6.

Answer: In this paper, the sensor response signal is tested only at the NO_2 concentration of 100, 200, 400 and 500 ppm, so each set of data has only 4 points which are insufficient. Each signal value is the average value after three tests of the same sample. It may be due to the experimental error and the insufficient samples, the relationship between the response signal and the logarithm of the NO_2 concentration does not have a good linear relationship, and this phenomenon is very common in the potentiometric sensor literature. Sensors of this structure generally follow the mixed-potential mechanism, so we usually use a linear fitting method when analyzing the potential signal with the logarithm of gas concentration. Therefore, we use the linear fitting method and R^2 to evaluate the degree of compliance of the linear relationship. Figures 6a, b, and c are test results for the operating temperatures of 500, 550 and 600 °C, respectively, so there are multiple Figs 6.

5) It is very unclear how (if at all) the circuit in Fig. 8 fits with Fig. 7. The authors are recommended to carry out DRT analysis (see <https://www.bio-logic.net/wp-content/uploads/AN60.pdf> and key references therein).

Answer: We are very grateful to the reviewer for the DRT analysis method file. This is a very effective analytical method that has been widely used in fuel cells and batteries, but it is rarely used in the field of gas sensor applications. We have used this tool to analyze my data and made changes in the manuscript. The analysis process using this method makes the mechanism explanation of the work more convincing

6) The quality of the figures needs improvement as the SEM is not aligned and the fonts are elongated. The schematics are visually unappealing. Also, the labeling of the figures should be formatted consistently.

Answer: Thanks for the reviewer's suggestion. This problem does exist, we have revised the manuscript.

7) The writing should be improved, and the article should be checked by a native

speaker or a professional editor.

Answer: We have already invited a researcher who is working in an English speaking country to review our manuscript, and the grammar and vocabulary errors have been modified according to your suggestion.